# CLUSTERING AND ENTITY MATCHING VIA LANGUAGE MODEL COMMUNITY DETECTION

## ABSTRACT

We introduce LMCD, a novel framework for semantic clustering and multi-set entity matching problems, in which we employ graph community detection algorithms to prune spurious edges from match graphs constructed using embedding and language models. We construct these match graphs by retrieving nearest embedding neighbors for each entity, then querying a language model to remove false positive pairs. Across a variety of cluster size distributions, and for tasks ranging from sentiment and topic categorization to deduplication of product databases, our approach outperforms existing methods without requiring any finetuning or labeled data beyond few-shot examples, and without needing to select the desired number of clusters in advance. Our embedding and inference stages are fully parallelizable, with query and computational costs which scale near-linearly in the number of entities. Our post-processing stage is bottlenecked only by the runtime of community detection algorithms on discrete graphs, which are often near-linear, with no explicit dependence on embedding dimension or numbers of clusters. This is in stark contrast to existing methods relying on high-dimensional clustering algorithms that are difficult to apply at scale; for entity matching our approach also ensures consistency constraints across matches regardless of group sizes, a desirable practical feature which is absent from all prior approaches other than vector clustering. Our improvements over previous techniques are most stark when clusters are numerous and heterogenously-sized, a regime which captures many clustering and matching problems of widespread practical importance.

## 1 INTRODUCTION

Modern language models have proven useful for a broad variety of "data wrangling" tasks, from data imputation and error detection (Narayan et al., 2022) to reference resolution (Moniz et al., 2024) log parsing (Xiao et al., 2024), named entity recognition (Wang et al., 2023), and pairwise entity matching (Peeters & Bizer, 2024). We study two such classes of problems — semantic clustering and (multi-set) entity matching — from a unifying perspective which leads us to consider LLM-based pipelines as end-to-end systems rather than granular decision engines. Indeed, as standard question-answer benchmarks have become increasingly saturated by frontier models (Taghanaki et al., 2024; McIntosh et al., 2024), it is also widely observed that LLM "agents" often struggle to complete complex multi-step tasks, particularly when the degree of sequential dependence is high (Chen et al., 2024; Han et al., 2024; Xing et al., 2024). Circuit complexity analysis has provided a partial explanation for this, showing that Transformers (Li et al., 2024b) cannot natively solve highly sequential tasks in zero-shot settings; further, periodic "hallucinations" of LLMs (Kalai & Vempala, 2024) may be inevitable if we hope to apply them to tasks which are ambiguous or out-of-distribution. As such, this motivates the careful design of *algorithms* for language model-based processing pipelines which are robust to these failure modes, yet which still produce outputs that "type-check" with the constraints of a given problem (i.e. a partition into disjoint sets).

We adopt this approach for semantic clustering and entity matching problems, employing language and embedding models to construct a "match graph" which can then be partitioned via community detection algorithms, avoiding several drawbacks of existing methods while also improving upon their experimental results. While these two tasks have historically been studied independently, with parallel sets of techniques, we note that they can both be viewed as instances of a general problem (which we refer to as "multiset matching"): there is a universe of $n$ entities, which must be parti-

tioned into disjoint subsets along a particular semantic axis. Typical modern approaches to semantic clustering involve applying high-dimensional clustering algorithms (e.g. KMeans(++) (MacQueen, 1967; Arthur & Vassilvitskii, 2007) or (H)DBSCAN (Ester et al., 1996; Campello et al., 2013)) to embedding vectors for each entity (Zhang et al., 2023; Viswanathan et al., 2023; Petukhova et al., 2024), generally targeting tasks where most clusters are large and the number of clusters is known (and small relative to the number of entities). In contrast, the bulk of recent literature on entity matching with language models (Li et al., 2020; Narayan et al., 2022; Peeters & Bizer, 2024; Wang et al., 2024) considers only the pairwise variant of the problem involving a semantic join between two databases, where a "blocking" stage (e.g. embedding or keyword similarity) first produces a set of candidate pairs to be matched. In fact, many standard benchmarks consist of "pre-blocked" pairs (Köpcke et al., 2010; Wang et al., 2021), where matching methods are then evaluated primarily via binary classification metrics. However, as noted by Peeters et al. (2023), the pairwise variant of the problem fails to capture many matching tasks of large practical importance, such as deduplication and linkage of parsed, scraped, or manually entered entities. The benchmark suite they introduce, WDC Products, contains several splits in which the ground-truth group sizes range from 2 to 10+. We focus primarily on this variant, as pairwise matching is largely "solved" by frontier LLMs with minimal prompting; yet, we are unaware of any prior methods for multiset matching which enforce group consistency constraints on outputs without relying on high-dimensional clustering. As such, we believe that our LMCD framework represents the most effective approach to date for addressing multi-set matching problems in practice; in addition to its simplicity of implementation, it also yields interpretable outputs in the form of a match graph, which may be further post-processed via human-in-the-loop approaches.

Our interest in avoiding explicit vector clustering is two-fold. Our first consideration is runtime scaling: when employing an embedding model with dimension $d$ for a set $n$ entities, running just a single step of $K$-means takes $O(n^2d)$ time when the number of groups is $O(n)$, whereas most instantiations of our framework run in time $\widetilde{O}(dn+n^2)$ [1] regardless of the number of groups. Our framework makes $O(n)$ queries to embedding and language models across three stages, each of which can be straightforwardly parallelized, after which we can "throw away" all embedding vectors, leaving only a size-$O(n)$ discrete graph for postprocessing via community detection. The second factor is with regards to the curse of dimensionality. Accurate clustering is widely observed to be difficult in high dimensions, as conventional principles related to distance and shape begin to break down (Radovanović et al., 2010; Peng et al., 2024). This creates a tension with leveraging benefits from state-of-the-art pretrained embedding models, which often have embedding dimensions as large as 4,096 (Xiao et al., 2023; Lee et al., 2024). This is exacerbated when the number of clusters is large and cluster sizes are non-uniform, as clustering algorithms such as KMeans (MacQueen, 1967) or HDBSCAN (Campello et al., 2013) may interpret small clusters as "noise", yielding poor performance. Despite this, modern data structures for approximate nearest neighbor search such as HNSW (Malkov & Yashunin, 2018) remain effective in many dimensions, and are widely used in performant vector databases for ranking and retrieval tasks (Lewis et al., 2021; Muennighoff et al., 2023) in conjunction with high-dimensional embeddings. Our approach demonstrates that nearest-neighbor queries are indeed *all* we need from embeddings in order to reconstruct high-fidelity matchings, even for heterogeneous cluster size distributions, when applying appropriate post-processing with language models and graph community detection algorithms.

Across our experiments, we find that pretrained embedding models in conjunction with vector databases are effective for efficient retrieval of candidate pairs with nontrivial precision, that and pretrained language models exhibit both high precision and recall when matching candidate pairs. Adopting these observations as primitives, our approach has a simple and intuitive interpretation. The "ground truth" match graph for a set of entities consists of disconnected components which are each cliques, and embedding-based retrieval yields a set of edges which is "dense" with respect to the true edges (though which may have many false positives). Language model queries then filter most of the false positives while retaining most of the true positives, yet some remaining false positive edges are inevitable. At this point, the "shape" of the ground truth match graph is sufficiently defined such that community detection algorithms can find cuts in the match graph which eliminate remaining false positive edges. Our results in Section 5 explore a number of tradeoffs related to dataset and cluster sizes, as well as algorithmic choices for community detection. Our method outperforms existing approaches for semantic clustering across a broad variety of datasets, and we

---

[1] $\widetilde{O}(\cdot)$ hides logarithmic factors.

also obtain competitive first-in-kind performance results for multi-set entity matching. Namely, we obtain the first multi-set matching results for the WDC Products datasets (Peeters et al., 2023) which enforce consistency constraints for transitivity of matches, where we also observe that application of vector clustering methods fails to surpass a trivial baseline.

We survey relevant prior methods for semantic clustering and community detection in Section 2, and discuss the datasets we consider in Section 3. We introduce the LMCD framework in Section 4 and present experimental results in Section 5. Our primary contributions are summarized as follows:

- We introduce the LMCD framework for multi-set matching, providing a unifying approach for semantic clustering and entity matching problems which obviates the need to perform explicit high-dimensional clustering.

- We demonstrate that LMCD improves quantitatively over prior methods for semantic clustering while exhibiting better runtime scaling and without needing to tune parameters for distance thresholds or cluster counts in advance.

- We show that LMCD also exhibits strong performance for multi-set entity matching, most notably in the sparse regime where clusters are abundant and small, where no existing methods have been effective under group output constraints.

## 2 BACKGROUND

### 2.1 CLUSTERING AND MATCHING VIA EMBEDDING SIMILARITY

Following a line of work in which pretrained embedding models (e.g. BERT (Devlin et al., 2019)) were finetuned for clustering and matching tasks via contrastive loss objectives (Shi & Wang, 2021; Li et al., 2020), requiring large amounts of labeled data for training purposes, more recent works (Zhang et al., 2023; Viswanathan et al., 2023; Petukhova et al., 2024) have begun incorporating self-supervised data augmentation, such as concatenation of LLM-extracted keyphrase embeddings or finetuning on LLM-generated labels, to supplant the dependence on labeled data. This issue is also partially addressed by employing "promptable" embedding models such as Instructor (Su et al., 2023), as is done by Zhang et al. (2023). We leverage this capability as well, via the bge-en-icl model from BAAI (Xiao et al., 2023); however, as we only use embedding vectors for nearest-neighbor retrieval rather than explicit clustering, we find that our methods are competitive without any finetuning, and we instead rely on few-shot prompting in our LLM-querying stage to inject precise distributional knowledge.

Though these prior methods most commonly employ Kmeans(++) (MacQueen, 1967; Arthur & Vassilvitskii, 2007) for vector clustering, (H)DBSCAN (Ester et al., 1996; Campello et al., 2013) presents a relevant alternative particularly when the number of clusters is unknown or expensive to tune. However, we find that retrieval and querying with modern models is sufficiently precise to render high-dimensional clustering unnecessary, as it can be substituted for much more computationally lightweight community detection algorithms running on discrete match graphs.

### 2.2 GRAPH COMMUNITY DETECTION ALGORITHMS

While originally developed for network analysis applications, community detection algorithms have found abundant applications across countless domains; see Li et al. (2024a) for a recent overview. Notably, community detection has proven useful for uncovering latent structure in knowledge graphs (Vahdati et al., 2018; Rollo & Po, 2022); our work falls within this line of inquiry.

While language models have been employed in conjunction with graph structures for retrieval and reasoning techniques Pan et al. (2023); Besta et al. (2024), our LMCD framework is the first approach to our knowledge which directly applies graph algorithms to the *outputs* of LLMs in order to filter and aggregate granular responses.

The appropriate choice of community detect algorithm choice for a given problem can depend on graph topology as well as computational considerations; we summarize the theoretical runtimes several community detection algorithms in Table 1, alongside popular algorithms for high-dimensional clustering. The listed runtimes are based on worst-case guarantees, with the exception of the Leiden

algorithm, whose oft-observed empirical runtime is essentially "folklore" (owing to its precessessor, the Louvain algorithm) (Traag et al., 2019). The number of iterations $H$ used by the Walktrap (Latapy & Pons, 2004) and Greedy Modularity Newman (2006) algorithms is at most $n$, and $O(\log n)$ iterations are often found to be sufficient in practice (with each corresponding to a "cut" in the graph). While the Girvan-Newman algorithm Girvan & Newman (2002) is oft-observed to generate high-quality partitions, it scales poorly to large graphs, and we omit it from our experimental analysis. In addition to the Leiden, Greedy Modularity, and Walktrap algorithms, we also analyze the Infomap algorithm (Rosvall & Bergstrom, 2008), which is inspired by compression of random walks. While Infomap lacks widely-known runtime guarantees, we find that its performance is similar to Walktrap in our experiments, and is practical to run on reasonably large graphs.

| | Clustering | | | Community Detection | | | |
|---|---|---|---|---|---|---|---|
| | $K$-Means++ | DBSCAN | HDBSCAN | GreedyMod | Leiden (obs.) | Walktrap | Girvan-Newman |
| Runtime | $O(ndKH)$ | $O(n^2d)$ | $O(n^2d)$ | $O(mH\log n)$ | $O(n\log n)$ | $O(mnH)$ | $O(m^2n)$ |

Table 1: Runtimes for clustering and community detection algorithms with $n$ entities.

## 3 TASKS

### 3.1 SEMANTIC CLUSTERING

We evaluate LMCD on 8 standardized datasets for semantic clustering, largely inheriting the methodology from the ClusterLLM experiments by Zhang et al. (2023). In Table 2 we survey the shape of each of dataset in terms of number of clusters as well as their range of sizes. We also present statistics on the outputs of the retrieval and matching stages of our framework (presented in Sections 4.1 and 4.2) after removing the 24 entities used as few-shot examples for each dataset. We retrieve $k = 10$ candidate matches for each entity via embedding nearest-neighbor queries, yielding sets of candidate pairs which are nearly half true positives (at least) for each dataset. We use a language model to query each candidate pair for validity, obtaining pairwise F1 scores above 90 for all but the GoEmo dataset[2].

| Dataset | Properties | | | | Retrieval ($k = 10$) | | | Matching ($p_{yes} > 0.5$) | | |
|---|---|---|---|---|---|---|---|---|---|---|
| | # Entities | # Clusters | (min, max) | # Few-Shot | # Pairs | # Pos | # Neg | Precision | Recall | F1 |
| Bank77 | 10,003 | 77 | (35, 187) | 24 | 99,790 | 91,761 | 8,029 | 92.58 | 97.59 | 95.02 |
| GoEmo | 23,485 | 27 | (39, 2710) | 24 | 234,610 | 99,410 | 140,200 | 43.56 | 84.25 | 57.43 |
| CLINC(I) | 15,000 | 150 | (100, 100) | 24 | 149,760 | 136,947 | 12,813 | 95.71 | 96.33 | 96.02 |
| CLINC(D) | 15,000 | 10 | (1000, 1000) | 24 | 149,760 | 143,633 | 6,127 | 97.08 | 99.14 | 98.10 |
| MTOP(I) | 15,638 | 102 | (1, 1616) | 24 | 156,140 | 137,861 | 18,279 | 91.99 | 92.49 | 92.24 |
| MTOP(D) | 15,677 | 11 | (929, 2187) | 24 | 156,530 | 150,555 | 5,975 | 97.13 | 98.80 | 97.96 |
| Massive(I) | 11,510 | 59 | (14, 810) | 24 | 114,860 | 90,699 | 24,161 | 85.86 | 94.83 | 90.12 |
| Massive(D) | 11,514 | 18 | (211, 1688) | 24 | 114,900 | 97,278 | 17,622 | 87.36 | 99.10 | 92.86 |

Table 2: Statistics for clustering datasets, retrieved pairs, and pairwise matching results.

### 3.2 MULTI-SET ENTITY MATCHING

To demonstrate the effectiveness of LMCD for multi-set entity matching with consistency constraints, we focus on the WDC Products collection of datasets consisting of web-scraped product listings, recently introduced by Peeters et al. (2023). The authors note that many standard entity matching benchmarks (Köpcke et al., 2010; Wang et al., 2021) only address pairwise matching, and thus fail to capture the additional challenge posed by aggregation of entities from more than two sources. The dataset is broken into several splits, ranging in the percentage of examples which have been identified as hard "corner cases" (20, 50, or 80). Still, the baseline approaches considerd by Peeters et al. (2023) involve formulating multi-set entity matching as multiclass classification without enforcing consistency constraints, and withhold a majority of the examples for finetuning.

---

[2]Throughout the paper we report precision, recall, F1, and NMI scores as normalized between 0 and 100.

To generate larger datasets with heterogenously sized clusters, we merge the train, validation, and test sets for each corner case split, and then again merge these into a single set, yielding four semi-overlapping splits with a broader range of cluster sizes. We withhold 24 entities from each split to be used for few-shot examples; as above, we report cluster statistics and results for retrieval and matching stages in Table 3.

| Dataset | Properties | | | | Retrieval ($k = 10$) | | | Matching ($p_{\text{yes}} > 0.5$) | | |
|---------|-----------|------------|------------|-------------|---------|--------|--------|-----------|--------|--------|
| | # Entities | # Clusters | (min, max) | # Few-Shot | # Pairs | # Pos | # Neg | Precision | Recall | F1 |
| WDC-cc20 | 6,020 | 1,000 | (2, 17) | 24 | 59,960 | 26,529 | 33,431 | 90.77 | 94.05 | 92.38 |
| WDC-cc50 | 6,044 | 1,000 | (2, 17) | 24 | 60,200 | 27,525 | 32,675 | 90.81 | 93.41 | 92.09 |
| WDC-cc80 | 6,056 | 1,000 | (2, 17) | 24 | 60,320 | 26,173 | 34,147 | 84.74 | 96.17 | 90.09 |
| WDC-all | 10,422 | 1,600 | (2, 27) | 24 | 103,980 | 43,256 | 60,724 | 90.62 | 90.94 | 90.78 |

Table 3: Statistics for WDC Products multi-set matching splits (Peeters et al., 2023), retrieved pairs, and pairwise matching results. Train, validation, and test sets are merged for each split.

## 4 COMMUNITY DETECTION WITH LANGUAGE MODELS

We present our Language Model Community Detection (LMCD) framework in Algorithm 1, with each stage described in greater detail below.

---
**Algorithm 1** LMCD

---
**Require:** Entities $\boldsymbol{X} = \{\boldsymbol{x}_i : i \in [n]\}$, parameter $k$, embedding model EMB$(\cdot)$, language model LLM$(\cdot)$ with prompt $\boldsymbol{p}$, community detection algorithm CD$(\cdot)$
1: Initialize vector database $\mathcal{B} = \{\boldsymbol{v}_i = \text{emb}(\boldsymbol{x}_i) : \boldsymbol{x}_i \in N\}$
2: Initialize graph $\mathcal{G}(\boldsymbol{X}, \boldsymbol{E})$ with $\boldsymbol{E} = \{\}$
3: **for** each $\boldsymbol{x}_i \in \boldsymbol{X}$ **do**
4:     Retrieve top $k$ similar vectors $\{\boldsymbol{v}_j\}$ from $\mathcal{B}$ using EMB$(\boldsymbol{x}_i)$
5:     **for** each retrieved $\boldsymbol{v}_j$ **do**
6:         $\boldsymbol{q}_{ij} \leftarrow$ LLM$(\boldsymbol{x}_i, \boldsymbol{x}_j; \boldsymbol{p})$
7:         **if** $\boldsymbol{q}_{ij} =$ "Yes" **then**
8:             Add edge $(\boldsymbol{x}_i, \boldsymbol{x}_j)$ to $E$
9:         **end if**
10:     **end for**
11: **end for**
12: $\mathcal{Z} \leftarrow$ CD$(\mathcal{G})$
13: **return** $\mathcal{Z}$

---

### 4.1 BLOCKING VIA EMBEDDING SIMILARITY

We begin by computing embedding vectors for string representations of each entity, using a pre-trained embedding model (namely, BAAI/bge-en-icl (Xiao et al., 2023)) in conjunction with a contextual prompt which illustrates the semantic focus for retrieval (see Appendix A for prompt and representation details). These vectors are then inserted into a vector database indexed by the Hierarchical Navigable Small Worlds (HNSW) data structure, which supports approximate $k$-nearest-neighbor retrieval in $O(\log n)$ time (Malkov & Yashunin, 2018).

For each entity, we then use HNSW to retrieve its $k$ nearest neighbors in terms of cosine similarity of embedding vectors, yielding a set of $nk$ "candidate matches" to be processed in subsequent stages. While the parameter $k$ can be tuned as desired, we find that selecting $k = 10$ is sufficient across all of our experiments to obtain strong results. The primary goal of this stage is to ensure that every entity is connected via candidate pairs to several others in its ground-truth group, yet the total degree of interconnection within each group may be sparse while still enabling a high-fidelity clustering (as long as it is sufficiently more dense than between-group candidate connections after the querying stage from Section 4.2 is applied). As such, selecting too large of a value for $k$ may present challenges in subsequent stages, as there may become a higher proportion of false positive connections which must be filtered downstream.

## 4.2 Graph Construction via Pairwise Queries

Given a set of candidate pairs, we construct a "match graph" by querying a language model (Llama-3.1-70B-Instruct in our experiments (Dubey et al., 2024)) as to whether each pair constitutes a true match, guided by a domain-specific prompt and a set of few-shot examples. The prompts we use are presented in Appendix A, and we discuss our procedure for choosing few-shot examples in Section 5.1. The intention of this stage, as illustrated by our numerical results in Tables 2 and 3, is to prune the bulk of the false positive candidate matches introduced by the previous stage, while retaining the bulk of the true positive pairs. When these two stages succeed, we are left with a graph which may not yet constitute a compelling partition of the entities into disjoint clusters, but where there are still clearly defined "communities" in terms of degree of interconnection.

## 4.3 Graph Partitioning via Community Detection

The final step of our approach is to simply apply a community detection algorithm (such as those discussed in Section 2.2) to our match graph, and return the generated partition as our clustering. Several community detection algorithms (notably, Walktrap) return *vertex dendrograms* (corresponding to sequences of cuts or merges) rather than only a single cut; while these algoriths return a default split according to numerical metrics, the easily interpretable nature of a sequence of splits (paired with text descriptions of entities) enables a straightforward interactive process for selecting a final partitioning of the entities. Further, given the broad variety of community detection algorithms with performant implementations, this provides a powerful toolkit for practitioners to experiment with various post-processing choices via inspection, removed from the computational burden of manipulating high-dimensional embedding vectors.

# 5 Experimental Results

## 5.1 Setup

For each entity, we retrieve the top $k = 10$ approximate nearest neighbors in embedding cosine similarity (excluding the entity itself) as candidate pairs, using HNSW (Malkov & Yashunin, 2018) via ChromaDB. Duplicate retrieved pairs are retained for the query stage, though each (undirected) edge is included at most once in our graph construction stage. For all experiments we use the bge-en-icl embedding model from BAAI (Xiao et al., 2023), an encoder-only model derived from Mistral-7B (Jiang et al., 2023) which led the MTEB Leaderboard (Muennighoff et al., 2023) at the time of our experiments, as well as Llama-3.1-70B-Instruct (Dubey et al., 2024) from Meta AI, hosted locally via vLLM (Kwon et al., 2023). We remark that the highly parallelizable nature of our approach is particularly amenable to leveraging prefix caching for prompts and few-shot examples (as is supported by vLLM). Additionally, we employ structured generation via Outlines (Willard & Louf, 2023) to ensure that queries only return either "Yes" or "No" as a response. While we include edges in our graph greedily according to maximum likelihood (e.g. if $\Pr[\text{Yes}] > 0.5$), we can also capture logprobs for each response to use for optional post-processing. Our prompts and representation formats are fairly generic, and similar to those from prior works (see Appendix A). We use the community detection implementations from the igraph library (Csardi & Nepusz, 2006).

To create few-shot examples, we select 6 clusters at random among those with at least 5 entities. From each of these clusters, we select two entities at random to form a positive example, and then a third as a negative example, paired with a random choice among the top 200 entities belonging to other clusters in embedding similarity. Each of these 24 example entities is then removed from the dataset. We did not substantially attempt to tune the number of examples or their selection strategy. We expect that further improvements could be obtained via more delicate example selection; however, our primary aim with LMCD is to demonstrate a method which can be applied to multi-set matching problems in a largely black-box fashion, without requiring model finetuning or manual optimization of prompts and hyperparameters. We note that this selection strategy can be efficiently approximated without access to any ground-truth labelings, via human inspection of top-$k$ retrieved nearest neighbors for a small number of randomly chosen entities.

## 5.2 CLUSTERING

In Table 4, across 8 clustering datasets, we find that LMCD surpasses or is competitive with all prior approaches, without requiring any finetuning or advance knowledge of cluster counts.

We compare against the leading methods from ClusterLLM (Zhang et al., 2023) and Viswanathan et al. (2023) (Keyphrase Clustering), as well as the other baseline methods (KMeans, SCCL, self-sup) presented by Zhang et al. (2023). We evaluate clusterings using normalized mutual information (NMI); while Zhang et al. (2023) also report multi-class accuracy numbers under optimal label permutations (which are largely correlated with NMI), this is less applicable when the number of clusters is not known in advance, or when the predicted and true cluster counts are allowed to differ (as is the case for LMCD).

We also experimented with applying KMeans (using the ground-truth $K$), DBSCAN, and HDBSCAN directly to the embedding vectors used by LMCD (from bge-en-icl). These results were uniformly worse than those found by the KMeans baselines from Zhang et al. (2023), and are omitted; we include the results for KMeans and HDBSCAN for our entity matching experiments in Section 5.3, where fewer relevant baselines are available, though DBSCAN failed to obtain scores nontrivially above zero across our tests. We expect the weaker performance of vector clustering on our bge-en-icl embeddings is in part due to their higher dimensionality (4,096 vs 1,024) as well as due to lack of finetuning.

We observe that the clustering datasets in which our approach shows the most substantial gains over prior results — Bank77, GoEmo, MTOP(I), and Massive(I) — typically have many clusters with a wide range of sizes (as seen in Table 2). In contrast, the datasets where we match or undershoot past results have fewer than 20 clusters of more uniform sizes, with the exception of CLINC(I) where clusters are identically sized.

We compare LMCD against applying community detection directly to candidate pairs, bypassing the LLM querying stage (EmbCD), and find compelling evidence for the value of this additional filtering stage. We also compare across several choices of graph algorithms, and find Walktrap (with 10-step walks) to be most effective. The final row of results in Table 4 showcase the optimal NMI values achievable when choosing Walktrap stopping times in terms of the ground truth labels. While this is not explicitly a fair comparison as it leverages true labels (and thus is omitted from our direct comparison in Table 4), it is perhaps indicative of the results which are easily attainable with additional manual postprocessing (though in most cases this gap is already small).

| Method | Intent Discovery | | | | Emotion | Domain Discovery | | | Average |
|---|---|---|---|---|---|---|---|---|---|
| | Bank77 | CLINC(I) | MTOP(I) | Massive(I) | GoEmo | CLINC(D) | MTOP(D) | Massive(D) | |
| KMeans* | 81.43 | 92.60 | 70.79 | 73.42 | 21.54 | 57.23 | 87.30 | 67.31 | 68.95 |
| SCCL* | 81.77 | 92.94 | 73.52 | 73.90 | 30.54 | 56.21 | 86.01 | 68.69 | 70.45 |
| self-sup* | 83.31 | 93.88 | 72.50 | 72.88 | 22.05 | 60.84 | 88.49 | 71.53 | 70.69 |
| FSC-Keyphrase | 82.40 | 92.60 | — | — | — | — | — | — | — |
| ClusterLLM-E-iter | 84.16 | 92.92 | 74.46 | 74.36 | 22.23 | 58.55 | 87.25 | 65.59 | 69.94 |
| ClusterLLM-I-iter | 85.15 | 94.00 | 73.83 | 77.64 | 23.89 | 54.81 | 89.23 | 68.67 | 70.90 |
| EmbCD-Infomap | 84.77 | 90.72 | 73.95 | 72.59 | 29.98 | 56.24 | 60.76 | 60.02 | 66.13 |
| EmbCD-W10 | 88.14 | 89.83 | 86.34 | 76.59 | 30.42 | 59.43 | 79.20 | 67.68 | 72.20 |
| LMCD-Greedy | 75.12 | 81.90 | 73.13 | 64.05 | 21.54 | 51.31 | 74.19 | 55.45 | 62.09 |
| LMCD-Leiden | 63.74 | 68.51 | 53.07 | 57.22 | 46.05 | 38.64 | 39.30 | 45.09 | 51.45 |
| LMCD-Infomap | 84.14 | 92.11 | 74.90 | 74.97 | 31.32 | 56.68 | 61.09 | 60.95 | 67.02 |
| LMCD-W5 | 88.26 | 92.63 | 87.53 | 78.90 | 29.08 | 60.56 | 79.35 | 66.58 | 72.86 |
| LMCD-W10 | 89.03 | 93.44 | 87.43 | 80.19 | 29.90 | 61.47 | 81.60 | 67.58 | 73.83 |
| LMCD-W15 | 89.01 | 92.81 | 87.59 | 78.84 | 30.52 | 61.22 | 81.18 | 68.17 | 73.67 |
| LMCD-W (opt) | 89.16 | 94.28 | 88.50 | 80.36 | 30.90 | 61.51 | 88.83 | 68.77 | 75.29 |

Table 4: Comparison of NMI evaluations across clustering datasets and methods. Averages over all datasets are shown in the final column. Best results are bolded and second-best are underlined. Methods with (*) denote per-dataset maximums over sub-methods (e.g. implementation choices, models used), and the final row shows results with optimal Walktrap parameters (#steps, iteration choice) using ground truth labels.

## 5.3 Entity Matching

Our results the WDC Products datasets are presented in Tables 6 (Micro, Macro, and Pairwise F1) and 5 (NMI). Table 5 considers the designation of each entity as a singleton cluster as a baseline; all LMCD methods surpass this baseline significantly across each dataset split, while all other considered methods fall short. Again in Table 6 we observe a particularly drastic gap between the performance of LMCD and other baselines, highlighting the difficulty of multi-set entity matching in this regime as well as the power of our approach. Here, in contrast to Section 5.2, the Infomap algorithm outperforms Walktrap by a small margin; changing the number of steps for Walktrap did not meaningfully change the final clusterings for each split. Here the Greedy Modularity is reasonably competitive, much moreso than for clustering; as this algorithm exhibits near-linear runtime scaling, this suggests that LMCD may be effective even for significantly larger datasets.

Our optimal Micro F1 results (for the best Walktrap stopping time) are also close to the best multi-class classification Micro F1 results reported by Peeters et al. (2023) — 93.03, 91.73, and 89.33 for the 20cc, 50cc, and 80cc splits, respectively — though this is far from an apples-to-apples comparison, as their method does not impose consistency constraints and requires substantial training data (which includes many distinct products that are also present in the test set).

| Method | WDC-cc20 | WDC-cc50 | WDC-cc80 | WDC-all | Average |
|---|---|---|---|---|---|
| | NMI | NMI | NMI | NMI | |
| Baseline | 86.91 | 86.86 | 86.81 | 87.46 | 87.01 |
| Emb-KMeans | 70.52 | 70.44 | 70.40 | 71.78 | 70.78 |
| Emb-HDBSCAN | 50.24 | 49.65 | 49.76 | 51.21 | 50.22 |
| EmbCD-Infomap | 84.02 | 85.11 | 84.46 | 84.29 | 84.47 |
| EmbCD-W5 | 73.02 | 75.14 | 75.70 | 72.42 | 74.07 |
| LMCD-Greedy | _95.33_ | _95.95_ | _94.42_ | _95.64_ | _95.33_ |
| LMCD-Infomap | **97.33** | **97.48** | **96.57** | **97.24** | **97.16** |
| LMCD-W5 | 95.18 | 95.90 | 94.27 | 95.62 | 95.24 |
| LMCD-W (opt) | 97.37 | 97.63 | 96.91 | 97.41 | 97.33 |

Table 5: Comparison of NMI scores across WDC Products splits and methods.

| Method | WDC-cc20 | | | WDC-cc50 | | | WDC-cc80 | | | WDC-all | | | Avg. |
|---|---|---|---|---|---|---|---|---|---|---|---|---|---|
| | Macro | Micro | Pair | Macro | Micro | Pair | Macro | Micro | Pair | Macro | Micro | Pair | |
| Emb-KMeans | 20.92 | 17.07 | 2.61 | 21.48 | 17.18 | 2.67 | 20.08 | 17.14 | 2.88 | 19.62 | 16.07 | 2.46 | 13.35 |
| Emb-HDBSCAN | 12.57 | 9.10 | 0.27 | 12.02 | 8.47 | 0.27 | 11.60 | 8.56 | 0.31 | 13.60 | 9.57 | 0.17 | 7.21 |
| EmbCD-Infomap | 26.11 | 31.60 | 18.51 | 26.24 | 31.73 | 16.21 | 23.26 | 27.74 | 19.45 | 23.32 | 27.40 | 17.13 | 24.06 |
| EmbCD-W5 | 11.48 | 14.08 | 10.14 | 11.69 | 14.24 | 7.99 | 11.91 | 14.70 | 10.85 | 10.68 | 12.46 | 7.85 | 11.51 |
| LMCD-Greedy | _81.49_ | _74.08_ | _60.20_ | _83.37_ | _76.54_ | _65.59_ | _75.65_ | _67.03_ | _54.27_ | _82.47_ | _75.57_ | _62.53_ | _71.57_ |
| LMCD-Infomap | **87.99** | **86.76** | **82.08** | **88.97** | **87.29** | **82.47** | **82.35** | **80.37** | **75.25** | **88.50** | **86.97** | **82.48** | **84.29** |
| LMCD-W5 | 81.32 | 73.42 | 58.30 | 83.30 | 76.30 | 64.88 | 75.47 | 66.53 | 52.48 | 82.43 | 75.42 | 62.14 | 71.00 |
| LMCD-W (opt) | 92.08 | 91.24 | 88.03 | 92.44 | 91.76 | 88.95 | 85.88 | 84.71 | 80.58 | 91.69 | 90.80 | 87.01 | 88.76 |

Table 6: Comparison of F1 scores across WDC Products splits and methods. Walktrap parameters for entries in the final row are selected by optimizing over NMI scores (see Table 5).

## 6 Conclusion

We have introduced LMCD, a novel framework for semantic clustering and multi-set entity matching, which offers improved performance over existing methods while simultaneously being easier to scale. LMCD is agnostic to the distribution of cluster sizes, exhibiting strong performance across regimes, and does not require finetuning or extensive advance parameter selection or finetuning. Our framework produces match graph artifacts which are easily interpretable and amenable to computationally lightweight postprocessing, via testing a suite of graph algorithms or interactively selecting cuts. For practical matching challenges involving large volumes of entities, with unknown demands for deduplication and linkage, our results present a powerful approach for efficiently extracting latent semantic structure, while sidestepping the curse of dimensionality in its entirety.

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

## A  APPENDIX

The prompts we use for embedding and language model queries are given in Tables 7 and 8, respectively. All clustering datasets we consider (2) feature only a single text column, which we use directly as the string representation for each entity. We represent each entry from WDC Products as:

```
{{title}} ({{brand}}) --- ${{price}} ({{priceCurrency}})
```

Language model prompts are given as system prompts using the standard chat template formatting for Llama-3.1-70B-Instruct; few-shot examples and queries are represented as:

```
A: {{str_A}}\nB: {{str_B}}
```

where `str_A` and `str_B` are the string representations for each entity in a query pair.

| Dataset | Embedding Prompt |
|---|---|
| Bank77 | Given a bank assistance request message, return other messages corresponding to the same type of request. |
| GoEmo | Given a message, return other messages corresponding to the same primary emotion. |
| CLINC(I) | Given a message, return other messages corresponding to the same kind of intent. |
| CLINC(D) | Given a message, return other messages corresponding to the same scenario domain. |
| MTOP(I) | Given a message, return other messages corresponding to the same kind of intent. |
| MTOP(D) | Given a message, return other messages corresponding to the same scenario domain. |
| Massive(I) | Given a message, return other messages corresponding to the same kind of intent. |
| Massive(D) | Given a message, return other messages corresponding to the same scenario domain. |
| WDC | Given a product listing, return other product listings corresponding to the same product. |

Table 7: Prompts for embedding retrieval using BAAI/bge-en-icl (Xiao et al., 2023).

| Dataset | LLM Prompt |
|---|---|
| Bank77 | Given two bank assistance request messages, determine if these messages correspond to the same category of request. Respond Yes or No. |
| GoEmo | Given a message, return other messages corresponding to the same primary emotion. Respond Yes or No. |
| CLINC(I) | Given two messages, determine if the messages correspond to the same kind of intent. Respond Yes or No. |
| CLINC(D) | Given a message, return other messages corresponding to the same scenario domain. Respond Yes or No. |
| MTOP(I) | Given two messages, determine if the messages correspond to the same kind of intent. Respond Yes or No. |
| MTOP(D) | Given two messages, determine if the messages correspond to the same scenario domain. Respond Yes or No. |
| Massive(I) | Given two messages, determine if the messages correspond to the same kind of intent. Respond Yes or No. |
| Massive(D) | Given two messages, determine if the messages correspond to the same scenario domain. Respond Yes or No. |
| WDC | Given two product listings, determine if the listings correspond to the same product. Respond Yes or No. |

Table 8: Prompts for embedding retrieval using Llama-3.1-70B-Instruct (Dubey et al., 2024).

