# OpenReview forum: "Clustering and Entity Matching via Language Model Community Detection"
_ICLR.cc/2025/Conference — ICLR 2025 Conference Withdrawn Submission_

### Official Review · Reviewer_fEV1 · 2024-11-03

**Soundness:** 2
**Presentation:** 2
**Contribution:** 2
**Rating:** 3
**Confidence:** 5

**Summary:**

The paper presents an algorithm called Language Model Community Detection (LMCD) to solve the tasks of semantic clustering and entity matching.

LMCD has three steps:

Step 1) Blocking via embedding similarity: LMCD uses BAAI/bge-en-icl pretrained model [1] to get embedding vectors for string representations of each entity. To do this, the user of LMCD needs to input contextual prompts. LMCD subsequently inputs the embedding vectors into a vector database indexed by the Hierarchical Navigable Small Worlds (HNSW) data structure [2] to get each entity's k nearest neighbors (as measured by cosine similarity between their embedding vectors).  k is a hyper parameter. The authors set it to 10.

Step 2) Graph construction via pairwise queries: LMCD builds a “match graph” by asking Llama-3.1-70B-Instruct [3] whether each pair is a true match. The user needs to enter domain-specific prompts and a set of few-shot examples for this step.

Step 3) Graph partitioning via community detection: LMCD uses the Walktrap algorithm [4], which returns vertex dendrograms.

The paper also presents empirical results showcasing how LMCD is fast and outperforms other methods.

References

[1] Shitao Xiao, Zheng Liu, Peitian Zhang, and Niklas Muennighoff. C-pack: Packaged resources to advance general chinese embedding, 2023.

[2] Yu. A. Malkov and D. A. Yashunin. Efficient and robust approximate nearest neighbor search using hierarchical navigable small world graphs, 2018. URL https://arxiv.org/abs/1603.09320.

[3] Abhimanyu Dubey, Abhinav Jauhri, Abhinav Pandey, and Meta AI Llama Team (500+ authors). The llama 3 herd of models, 2024. URL https://arxiv.org/abs/2407.21783.

[4] Matthieu Latapy and Pascal Pons. Computing communities in large networks using random walks, 2004. URL https://arxiv.org/abs/cond-mat/0412368.

**Strengths:**

LMCD puts together existing methods to solve semantic clustering and entity matching. The authors pay particular attention to how fast LMCD runs.

**Weaknesses:**

Although LMCD appears to be a well-engineered algorithm, there are no guarantees for its performance. For example, community detection suffers from the no-free-lunch theorem (see https://arxiv.org/pdf/1903.10092). Not to mention all the problems associated with prompt engineering.

**Questions:**

1. Are there patterns in LMCD's errors?
2. How sensitive is LMCD to the prompt selection?

---

### Official Review · Reviewer_Kpmf · 2024-11-03

**Soundness:** 1
**Presentation:** 2
**Contribution:** 1
**Rating:** 3
**Confidence:** 4

**Summary:**

This paper studies clustering and entity matching. The paper develops several techniques based on language model and community detection. The experiments are conducted to evaluate the performance.

**Strengths:**

1.	The clarify of the paper is good. The presentation is easy to follow and straightforward.
2.	Clustering and entity matching are two important research problems well-studied in the literature.

**Weaknesses:**

1.	The paper lists several techniques, but the focus and technical contribution of this paper is unclear. For example, the paper just uses LM to build a graph, and then applies community detection techniques that are from existing work.
2.	As shown in Algorithm 1 of the paper, the LMCD is just a plain workflow to use LLM and CD methods. The novelty of the work is limited.
3.	In experiments, the performance of clustering is evaluated. However,  the paper uses community detection as a way to solve clustering, which is confusing. Community detection and clustering are basically quite similar to each other.
4.	In experiments, Table 4 shows many variants of the proposed LMCD. It is questionable if the proposed method can perform in a stable way. As shown in Table 4, the variants of LMCD have diverse performance.

**Questions:**

1.	What is the significance of the different settings in the variants of LMCD? Are they crucial to the performance of LMCD?
2.	What are the parameter settings of the experiments?

---

### Official Review · Reviewer_8SJk · 2024-11-03

**Soundness:** 2
**Presentation:** 2
**Contribution:** 2
**Rating:** 3
**Confidence:** 3

**Summary:**

This paper introduces a new method to solve semantic clustering and multi-set entity matching problems. The method consists of the following three steps. First, a KNN graph is (implicitly) constructed based on the vector embeddings of entities. Second, the edges of the KNN graph are filtered/refined as follows. For each edge of the KNN graph, if an LLM predicts that the edge is positive, i.e., if the LLM deems that the two entities should be assigned to the same cluster, then that edge is kept, otherwise, the edge is removed. Finally, a community detection algorithm is applied to the refined KNN graph to produce the final outputs. The authors demonstrate empirically that the proposed method outperforms existing methods for both semantic clustering and multi-set entity matching problems.

**Strengths:**

The proposed method is simple and intuitive. The empirical results are promising.

**Weaknesses:**

- Overall, as a methodology paper, there is very limited novelty in terms of method development. The proposed framework is simply applying  a community detection algorithm to a KNN graph whose edges are refined by LLM predictions. I think that the paper in its current form may be better suited for an applied data science venue.

- In addition, there is very limited analysis on the impact of individual components that make up the overall framework. For example, how different embedding methods affect the end result, how different LLMs affect the end result, how different ways of selecting the few-shot examples affect the end result, etc.

- The submission does not contain supplementary code to reproduce the empirical results.

**Questions:**

N/A

---

### Official Review · Reviewer_ju5R · 2024-11-06

**Soundness:** 3
**Presentation:** 3
**Contribution:** 2
**Rating:** 5
**Confidence:** 3

**Summary:**

Authors tackle the entity matching problem over multiple datasets via LLM and graph community detection algorithm. First, by using the LLM embeddings, authors put them into the vector database so that candidates are generated for each entity through nearest neighbor search like HNSW. Second, by using the LLM prompt engineering with few shot examples, authors produce positive matches among the induced candidates. Finally, authors perform the community detection algorithm to partition the graph for final matching outcomes.

For the experiments, authors perform on clustering tasks as well as the multi-dataset entity matching tasks. Authors show that the performance from the proposed method is better than baseline methods for both tasks.

**Strengths:**

- A straightforward and simple algorithm is proposed for the entity matching problem.
- Authors show the better performance of the proposed method over the baseline.
- Authors propose the framework so that some components are pluggable with any algorithm.

**Weaknesses:**

- The contribution to the ICRL community is marginal. Using the semantic embeddings and applying community detection algorithm for clustering tasks is well-known. LLM prompt part has been added, but then more contribution to make better prompt would be expected.
- From the entity matching task, most contributions seem to come from LLM prompt. As pointed out above, if that is the major component to determine the performance, it needs to be studied further.
- It is not easy to understand the impact of graph community detection algorithm for the given tasks. Particularly for the entity matching task where LLM prompt makes major contributions, the corresponding ablation study is needed.

**Questions:**

- Different clustering algorithms seem to work better for different datasets / tasks. How can we pick up the best algorithm when we do not know the entire ground truth?

- By comparing Table 5 and Table 6, NMI does not represent the actual accuracy of entity matching well. However, the clustering algorithm tries to optimize more on the NMI. How can the authors convince that better clustering algorithm would improve the entity matching accuracy?

- It would be great if authors have the ablation study that relies more on LLM prompt than graph clustering algorithms.

---

### Note · Authors · 2024-12-02

I have read and agree with the venue's withdrawal policy on behalf of myself and my co-authors.